# Application of Date (*Phoenix dactylifera* L.) Fruit in the Composition of a Novel Snack Bar

**DOI:** 10.3390/foods10050918

**Published:** 2021-04-22

**Authors:** Salam A. Ibrahim, Hafize Fidan, Sulaiman O. Aljaloud, Stanko Stankov, Galin Ivanov

**Affiliations:** 1Food Microbiology and Biotechnology Laboratory, North Carolina Agricultural and Technical State University, Greensboro, NC 27411, USA; ibrah001@ncat.edu; 2Department of Nutrition and Tourism, University of Food Technologies, 4002 Plovdiv, Bulgaria; docstankov@gmail.com; 3Department of Exercise Physiology, King Saud University, Riyadh 11451, Saudi Arabia; saljaloud@ksu.edu.sa; 4Department of Technology of Milk and Dairy Products, University of Food Technologies, 4002 Plovdiv, Bulgaria; ivanovgalin.uft@gmail.com

**Keywords:** date fruit, energy bars, paste, sensory, texture

## Abstract

The aim of the present study was to evaluate the possibilities for using date fruit from the Kingdom of Saudi Arabia in the formulation of a novel snack bar while replacing the used honey with date paste. The technological, textural, microbiological and sensory qualities of the obtained food products were evaluated during storage for 12 days after their production. Date palm fruit in the form of date paste was used as an ingredient in the composition of the new snack bars that also included nuts and dried fruits. Five formulations were prepared: a control bar, snack bar with 40% date paste, snack bar with 50% date paste, snack bar with 60% date paste, and snack bar with 70% dates paste. The resulting date paste’s textural characteristics supported the bonding potential of the food system and gave a sweet taste to the final product. The formulations containing 50% date paste presented the highest overall acceptability and were the formulation with the best sensory characteristics. Thus, the addition of date paste in snack bars would be a good option to develop a functional product that contributes to rational nutrition principles. The obtained values in the current study confirmed the technological and functional potential of date fruit as a product that can find adequate application in the composition of foods with functional properties.

## 1. Introduction

Changing eating habits and increased consumer demand for healthy foods with functional components increased the interest in improving many well-known food products’ nutritional characteristics. Today, the words “healthy or functional food” have become mandatory criteria for many consumers when they purchase food. The increased cost of treating diseases, greater life expectancy, the increasing number of older people in society, and the striving to improve the quality of life have all led to an increased demand for foods that meet people’s higher expectations. Flavorful food with excellent organoleptic characteristics is no longer adequate as it must also be nutritious. Consequently, the development of new food groups with specific textural, sensory, or functional properties has been the subject of several recent studies [1]. Moreover, the modeling of food formulations with a view to their marketing application in the diets of targeted cohorts of consumers has resulted in a significant impact on the economic feasibility of food products. For example, snack bars are popular with people around the world, and product preferences are often linked to health-related claims, package attributes, price and flavor [2]. Energy bars are an excellent option for breakfast or snacks after a sports workout, too [3]. The variety of nuts, seeds, soluble, and insoluble dietary fiber in the bars’ composition allows their easy adaptation to a rational diet as a source of slowly digestible carbohydrates [4], micro-and macronutrients. The inclusion of natural sweeteners, such as honey, dried fruits, and fruit pastes, increased the energy value of bars. On the other hand, it influenced their production’s technological characteristics and conditions [4,5]. Differences in the composition of the individual ingredients changed the nutritional matrix, creating conditions for change in the product’s nutritional, microbiological, textural, and sensory parameters [6].

However, the conventional snack bars sold in retail chain outlets are typically made with honey or sugar and glucose syrups. Since honey can be an allergen, food product developers have been looking at ways to replace honey with other non-allergenic ingredients. In addition to the fact that bee products show allergic reactions in the body [5], they have a high purchase price, which increases the financial evaluation of the final product. Alternative natural substitutes with technological qualities are fruit pastes, such as that of date palm fruits. Dates fit this requirement and can thus be used as an acceptable alternative to these two food raw materials [7]. The date palm is a tree with significant economic and nutritional value. The fruit from the date palm consists of date pulp and date seed, representing 10–15% of the whole fruit. Date seeds are a secondary product after utilization of the date pulp and are primarily used for animal feed in addition to application in food, cosmetic, and biotechnological industries (Figure 1). Date fruit has a low glycemic index and is a source of antioxidants, carbohydrates, dietary fibers, proteins, minerals and vitamin B complexes such as thiamine (B1), riboflavin (B2), niacin (B3), pantothenic (B5), pyridoxine (B6), and folate (B9). Carbohydrates comprise 70% of the date fruit mainly as fructose and glucose. The minerals in date fruit are calcium, iron, magnesium, selenium, copper, phosphorus, potassium, zinc, sulfur, cobalt, fluorine, and manganese [1]. A new approach in food technology is the development of new food products based on the application of low-temperature treatments or the absence of heat treatment. This new model necessitates studies related to assessing the chemical composition, nutritional value, texture, sensory evaluation, microbiological stability, etc., in order to demonstrate the effectiveness of the methods [8].

The aim of the present study was thus to utilize date fruit from the Kingdom of Saudi Arabia in the formulation of novel snack bars while replacing honey with date paste and to evaluate the bars’ technological textural, sensory, and microbiological parameters during storage. Our findings may help consumers make a healthy food choice for their snacks and also help regulatory authorities to select foods that could meet consumers’ nutritional needs.

## 2. Materials and Methods

### 2.1. Raw Materials Preparation

Date (*Phoenix dactylifera* L.) samples were provided by a National Organic Date company (Medjool brand) in Riyadh, the Kingdom of Saudi Arabia), and standard raw materials such as honey (Region Strandzha, Kosti village, Bulgaria) cashews, oatmeal, and dried cherries (Vesta Trading LTD, Plovdiv, Bulgaria) used in the current study were purchased from a local store in Plovdiv, Bulgaria and authorized by the Ministry of Health in Bulgaria.

### 2.2. Preparation of Date Bar’s Formulation

The samples were obtained in laboratory conditions in the technological laboratory of the University of Food Technology—Plovdiv, Bulgaria. The pulp of dates and dried cherries were separated from the seeds. The dates were ground into paste using a laboratory mill (Clatronic KSW 3307 Grinder) for 2–3 min. The other components—cashews and dried cherries are subjected to separate grinding with the mill. The oatmeal was used whole. The obtained date paste was then thoroughly mixed with other ingredients (oatmeals, dried cherries and cashews) in order to obtain uniform distribution. After mixing, sheeting was performed in order to create bars of 2.5 cm width, 1 cm height, and 7 cm in length. The sheeting was made using a metal roller and polyethylene foil. The control sample was obtained with polyfloral honey; as in the other samples, it was replaced with date paste. The date bars are cooled (4 ± 6 °C, for 30 min) in order to increase their hardness after being reduced due to the increased temperature from the mechanical impact of mixing. After being refrigerated, each date bar weighed approximately 25 g and they were packed individually with polyethylene foil and plastic packaging. The samples thus prepared and coded were stored for 24 h at room temperature (18 ± 2 °C) before their physical and sensory testing. The formulations of four date bars (Figure 2) containing date paste, cashews, dried cherries and oatmeal are given in Table 1, with ingredient amounts being shown as percentages. The control sample formulation was prepared according to traditional techniques as the ingredients (date paste or honey, oatmeal, dried cherries and cashews) were mixed.

The recipe composition of the analyzed samples represented the possibility for percentage replacement of polyfloral honey with date paste. Four different models were used with different participation of ingredients (with increasing participation of date paste) in order to obtain the typical texture of the bars.

### 2.3. Water Activity of the Snack Bars with Date Paste

The water activity (aw) of the obtained snack bar samples was measured at a constant temperature of 25 °C with a Novasina AG Neuheimstrasse 12 CH-8853 Lachen, Switzerland. The samples were put into a plastic cylinder with a height of 1 cm and a diameter of 4 cm which was then placed in the apparatus’s measuring chamber, the sensor was calibrated and readings were taken.

### 2.4. The Instrumental Texture of Snack Bars with Date Paste

The texture profile of the snack bars was evaluated using a Stablemicrosystems TAXT2 texture analyzer. The sample bars were twice compressed by a 10 mm diameter probe. Analyses were carried out in five replications for each sample. This method was used in order to determine hardness parameters such as rupture stress, [kPa].

### 2.5. Nutrition Information

The snack bar nutrition information was determined by the calculation method, which involves using snack bar formulation and raw ingredient nutrient data [9]. The nutritional value of the finished product was calculated per 100 g based on raw material specifications obtained from suppliers of each of the ingredients.

### 2.6. Microbiological Analyses

The microbiological analysis of the bars was carried out along with analyses for total plate count (TPC) [10], molds and yeasts [11], faecal coliforms [12], *Salmonella* species [13], coagulase-positive Staphylococci [14]. In addition, the following parameters were determined: total plate count (TPC) in 1 g of bar sample, total number of molds and yeasts in 1 g of bar sample, coliforms in 1 g of bar sample, *Salmonella* spp. in 25 g of bar sample and coagulase-positive staphylococci in 1 g of bar sample.

### 2.7. Sensorial Evaluation

The hedonic test for sensory profiling was used to establish the sensory characteristics (overall acceptability, appearance, flavour, sweetness and texture) of the snack bars, 24 h after production. The sensory profiling involved 90 panelists, who were homogeneously distributed by age and gender. They were students aged between 19 and 24 (male 47 and female 43), previously acquainted with the aim of the study. The sensory analysis was conducted in the specialized laboratory for consumer quality assessment at the University of Food Technology—Plovdiv, Department of Nutrition and Tourism. Samples of four different snack bars were kept in coded plates covered with aluminum foil. A nine-point hedonic scale was used to determine the degree of liking for snack bar products (9 = like extremely, 5 = neither like nor dislike, 1 = dislike extremely). Following a 30-min orientation session, the panelists began to record each sensory characteristic’s intensity and terminology and anchor points on the scale were specified. The coded samples were shown simultaneously and evaluated in random order among the panelists.

### 2.8. Statistical Analysis

The data were analyzed and presented as mean values ± standard deviation. Statistical analysis was carried out using Excel software. A one-way analysis of variance (ANOVA) was performed, and significant differences between samples were determined applying the Tukey’s honest significant difference (Tukey “HSD”) test, which is used to test differences among sample means for significance, at *p* < 0.05.

## 3. Results and Discussion

### 3.1. Water Activity and Hardness of Snack Bars Supplemented with Different Concentrations of Date Paste during Storage

Water activity (aw) is an important indicator that characterizes the content of free water in the system, which is necessary for microbial growth. While it is a measure of the availability of water molecules to enter into microbial, enzymatic, or chemical reactions, therefore its values can predict both the microbiological stability of the product and some textural indicators.

The aw values of five snack bar samples are presented in Figure 3.

The highest water activity aw (ranged between 0.869 and 0.889 during the storage period) was observed in the control sample formulation. The lowest aw (between 0.654 and 0.721) contained in a snack bar formulation had the lowest amount of date paste (40%). An increasing trend was observed in the formulations.

The increase in the date paste concentration in snack bar samples changes the level of water activity. The lowest values of aw were observed at the time of production in the sample with 40% date paste. However, the aw values among the other samples with date paste were closer together, and the highest value was noted in the control sample (0.869). The sample with 40% date paste (0.721) had the lowest aw after 12 days of storage. The relatively low water activity values in the sample with 40% date paste were maintained throughout the test period. The control sample shows a tendency to maintain high water activity, which can be a severe prerequisite for the emergence of microbiological processes or a change in the quality of the bars. Moreover, the control sample aw during storage increased the least (2%), while the values of aw of the sample with 50% date paste increased the most (9.7%). The values for aw and moisture in the obtained samples confirmed other authors’ results [15] related to changes in aw during storage. For example, the aw values for cereal bars ranged from 0.66 to 0.72 [16], while the water activity of muesli, pumpkin, and coconut bars varied from 0.63 to 0.74) [17]. Padmashree et al. [15] studied the aw values of composite cereal bars with regard to moisture levels, peroxide values and free fatty acid values for 45 days. They reported that the composite cereal bar equilibrated to 4.2, 8.0 and 10.6% moisture levels at 0.33, 0.57 and 0.73 water activities, respectively and the peroxide value and free fatty acid values were found to be lowest at 0.33 aw as compared to 0.57 and 0.73 aw. The free water level in the snack bar formulations with date paste was found to be lower than that of the control sample. The relationship between the level of aw and the sample humidity levels should also be considered as these two parameters are highly proportional [18]. The differences in the results obtained in the current study and those reported in the literature may be due to the different formulations and methodologies used to manufacture the cereal bars, among other variables.

### 3.2. The Instrumental Texture of Snack Bars with Date Paste

A general technological problem for snack bars is the appearance of stickiness formed by the participation of soluble sugars. It affects the textural and sensory performance of the product, giving it a more fluid and sticky structure. In industrial production, it is crucial to evaluate the effect of storage, during which some soluble fractions will change the texture. The changes in the hardness of snack bar samples during storage are presented in Figure 4. The development of hard texture during storage is the limiting factor for snack bars quality, although the product remained microbiologically safe and stable during this time. According to the results, the hardness of snack bars increased after 12 days.

Textural analysis showed that the lowest hardness (4.08 ± 0.31 kPa) occurred in the control sample. On the first day of storage, the hardness of snack bar samples comprised of 50% date paste (5.41 ± 0.24 kPa) and 60% date paste (5.55 ± 0.11 kPa) was found to be close. Similar hardness values were also reported for the samples with 40% date paste (5.98 ± 0.30 kPa) and the sample with 70% date paste (5.86 ± 0.12 kPa). It was also noted that the value for force used to tear the snack bar sample with 40% date paste was comparable to the value of force to tear the snack bar sample with 70% date paste. The change in hardness values of snack bar samples during storage has been reported in several studies [4,16]. The control sample in the current study was characterized by the lowest deviations from the initial hardness (0.42 ± 0.01 kPa), followed by the sample with a 40% date paste (1.03 ± 0.03 kPa). The hardness deviations were comparable for the samples with 40% and 50% date paste (1.61 ± 0.06 kPa) and the one with 70% (1.64 ± 0.02 kPa) date paste. The greatest changes in hardness values were reported for the sample with 60% date paste (2.52 ± 0.00 kPa).

The most pronounced increase in the hardness of the obtained snack bar samples was observed between the sixth and twelfth day of storage. The increase in the hardness curve moved most evenly in the control sample and the sample with 40% date paste. The results of the obtained analysis confirmed that dates could be used as an ingredient in order to retain moisture during the storage of such food products. The hardness values as a textural indicator confirmed previous analyses in which it was found that during storage, the hardness of the bars increased [4]. Damasceno et al. [16] performed an instrumental texture of cereal bars containing pineapple peel flour and reported that the compressive force increased significantly with the addition of 9% pineapple peel flour. The authors revealed that the optimum level of addition of pineapple peel flour is up to 6%. However, since an increase in bar consistency was observed with increasing concentrations of pineapple peel flour, no differences were observed with up to 6% flour addition compared to the control.

### 3.3. Nutritional Composition of Snack Bars Supplemented with Different Concentrations of Date Paste

Food products provide the body with the necessary nutrients to meet its energy, building, and regulatory needs. Different types of products have different chemical compositions, various nutritional value, and thus fulfill different nutrient needs. Moreover, we could say that the efficacy or usefulness of a food product is characterized by the product’s nutritional value as determined by the type and amount of nutrients in the food and the level of their absorption by the body. The nutritional value also includes the energy value of the food which reflects the amount of energy released in the body during the biological oxidation of macronutrients. Data on the nutritional value of foods can be used to determine their nutritional density.

The information on the nutritional composition of the snack bars with date paste is shown in Table 2.

The differences in the calculated quantitative composition are due to the differences in the recipe composition of the bars. The values of the studied samples were close to the previously reported ones [17,18]. The protein composition of the date bars in this study, were higher compared to the protein values of snack bars in the local market (4.4%).

Values for the fat content of snack bars in a much wider range have been reported by other authors [18,19,20,21], according to whom lipids varied between 1.28 to 17.43%, with the fat content depending on the raw materials used and their quantitative participation.

The amount of total carbohydrates reported in this study was close to the previously determined (varying between 62.93% and 80.08%) by Mourão et al. [20], as their quantitative composition depended on the raw materials involved. The calculated energy density values of date bars obtained in this study are close to the averages for the local trade market’s bars (400–450 kcal).

Nutritional information is thus data pertaining to nutrients and their values that are present in a food product and which helps consumers to make informed food choices. Moreover, nutritional claims related to the content of certain nutrients or other substances in the food product are often made. For example, according to the information reported in the regulation (EC)1924/2006 [22] of the European Parliament and of the Council of 20 December 2006 on nutrition and health claims made on foods, a claim that a food is a “source of fiber” may only be made if the product contains at least 3 g of fiber per 100 g whereas a food may have a claim that is “high in fiber” when the product contains at least 6 g of fiber per 100 g or at least 3 g of fiber per 100 kcal. Therefore, since the fiber content of the snack bars in the present study varied between 6.88 and 7.44 g, these bars would be considered as a significant source of fiber. On the other hand, a claim that a food is a source of vitamins and minerals may only be made if the product contains 15% of the nutrient reference values (NRV)/100 g. Thus, the snack bars made with date paste could be considered as a source of thiamin (values ranged between 0,16 and 0.26 mg), calcium (the amount varied between 102.10 and 144.55 mg), magnesium (151.90 and 153.80 mg) and zinc (1.18 and 2.07 mg) [23,24,25].

In a balanced diet, muesli bars can offer a convenient and nutritious snack, with many bars providing around 30% of an adult’s and up to half of a child’s daily requirement for whole grain, and more than half of all products are a source of fiber. Szydłowska et al. [23] developed high-protein bars using organic ingredients and reported that the novel products were found to be a good source of protein and fiber. The muesli and pumpkin bar with cocoa samples contained >10 g of fiber/100 g. The muesli and coconut bars contained over 20 g/100 g total protein while the pumpkin bars contained 17.3–19.1 g/100 g of protein.

### 3.4. Microbiological Analysis

Snack bar samples supplemented with different date paste concentrations were examined for microbial analyses over different storage periods (within twelve days of production at a temperature of up to 18 °C and relative humidity φ < 75%). In addition to the sensory quality, textural characteristics and nutritional value of the bars, microbiological quality is essential for consumer health and safety. The bars were thus evaluated in accordance with microbiological quality parameters before the sensory tests were conducted.

No pathogenic bacteria—coagulase-positive staphylococci in 1 g, *Salmonella* spp. in 25 g and faecal coliforms in 1 g, respectively, were detected. In addition, up to the 12th day of storage at room temperature, no visible mold was detected. At the end of the storage period, the microbial loads in the control and the variants were close. This was most likely due to secondary air contamination on the samples’ surfaces.

### 3.5. Sensorial Evaluation of Snack Bars, Supplemented with Different Concentration of Date Paste

A sensory analysis evaluation was performed in order to determine optimum sensory characteristics of snack bars in terms of panelist preferences. The results of the sensory evaluation are given in Table 3.

The rating test showed that the sensory characteristics of overall acceptability, appearance, flavor, sweetness and texture are perceived without significant differences between the control sample and supplemented with different concentrations of date paste snack bars. The ANOVA applied to the appearance, flavor, sweetness and texture data determined significant differences (*p* < 0.05). Our results showed that the snack bar with 50% date paste received the highest overall acceptability score from the panelists. In terms of appearance, the control sample and snack bars with the addition of 40% date paste no significant differences were observed, while the snack bar sample with 50% date paste was perceived to be the best. The data showed no great difference in the flavor values for the investigated snack bars with 60% and 70% date paste. The control sample exhibited a light yellow color, while the other snack bars had a medium to the dark surface due to the presence of date paste. With regard to the samples’ sweetness and texture, the bar with the addition of 50% date paste was rated the highest while all other samples received close scores.

A number of researchers have studied the effects of different ingredients on the sensory characteristics and overall perception of snack bars [16,25,26,27]. For example, Santos et al. [28] studied the sensory characteristics of homemade and alternative cereal bars using dehydrated jackfruit and seed meal as a fiber source. With regard to sensorial characteristics, formulations containing 30% and 40% of seed meal of jackfruit were preferred by testers. Damasceno et al. [16] developed four formulations of cereal bars containing pineapple peel flour. The samples scored 7.0 (moderately good) and 9.0 (very good) for hedonic parameters. Padmashree et al. [15] reported that a gradual but significant (*p* ≤ 0.05) decrease in all the sensory parameters (color, aroma, taste, texture and overall acceptability) occurred during storage. For example, the texture of the bars became harder during storage, negatively influencing the overall acceptability scores. According to Rios et al. [29], the acceptance of snack bars depended primarily on attributes such as “sweetness” and “good appearance”, and rejection occurred in formulations containing ingredients with high lipid content treated at high temperatures for a long time.

## 4. Conclusions

The results from the current study demonstrated that date paste can be used in the development of nutrition bars as healthy snacks. The date paste improved the obtained product’s technological qualities and did not create conditions for deterioration of the consumer assessment. When using 50% of the total mass, the fruits of the date palm improve the bars’ textural, sensory and technological qualities. More in-depth analyses on more extended storage would make it possible to create technological models to assess the impact on product quality indicators. A technological model was created to improve the textural and sensory parameters of snack bars obtained with the addition of date paste. The addition of dates had a positive effect on the bars’ nutritional status by increasing the food’s nutrient density. Moreover, the snack bars with date paste exhibited improvement over the control bars with regard to appearance, flavor, sweetness and texture, confirming that replacement with date paste had enhanced the sensory characteristics of the final products. The snack bar with 50% date paste received the highest overall acceptability rating, whereas microbiological studies suggested that storage of the snack bars did not lead to the development of pathogenic bacteria. Our results from this work also illustrated that the addition of date paste improved the textural characteristics (hardness) of the snack bars due to maintaining the system’s water balance. Despite the fact that there are some limitations regarding the shelf life and the necessity for comprehensive sensory analysis of the resultant bars, this paper could be the basis for the application of food sources with functional properties in the composition of novel snack bars. Further investigations regarding the food system’s microbiological stability and textural properties could be estimated for a more extended storage period. Our study’s findings may help consumers make the right choice of food for their snack and to school managements to select a set of foods that could meet students’ developmental needs.

## Figures and Tables

**Figure 1 foods-10-00918-f001:**
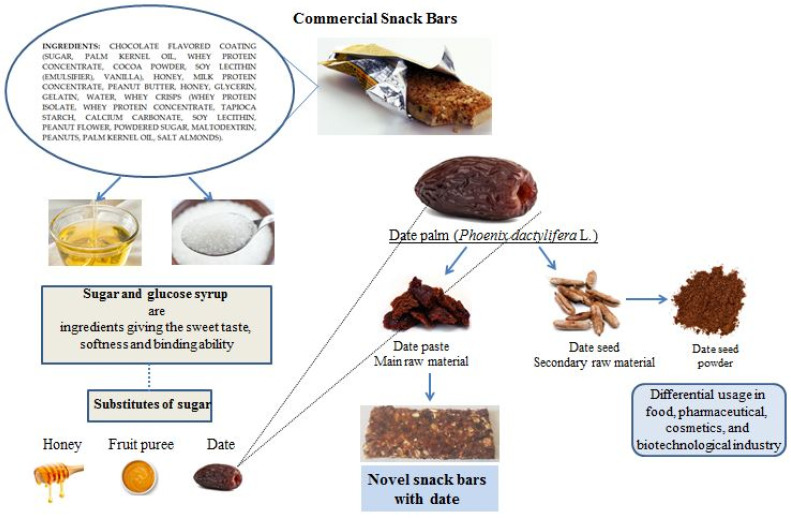
Production of novel snack bars with date palm.

**Figure 2 foods-10-00918-f002:**
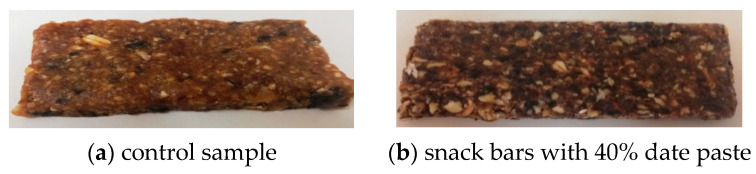
Snack bars with (**a**) control sample; (**b**) 40% date paste; (**c**) 50% date paste; (**d**) 60% date paste; (**e**) 70% date paste.

**Figure 3 foods-10-00918-f003:**
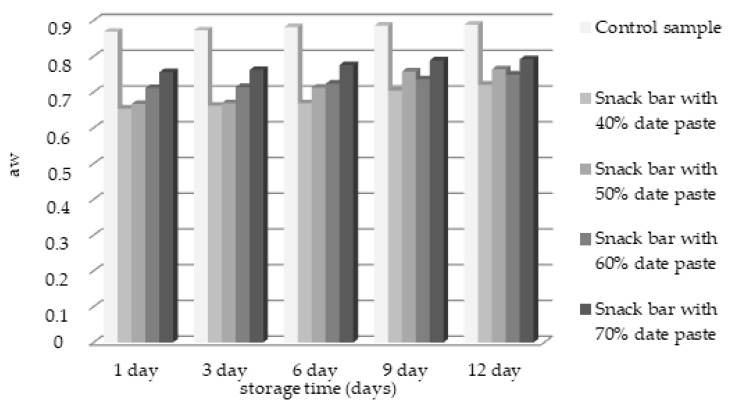
Changes in the water activity values of five snack bar samples during 12 days of storage.

**Figure 4 foods-10-00918-f004:**
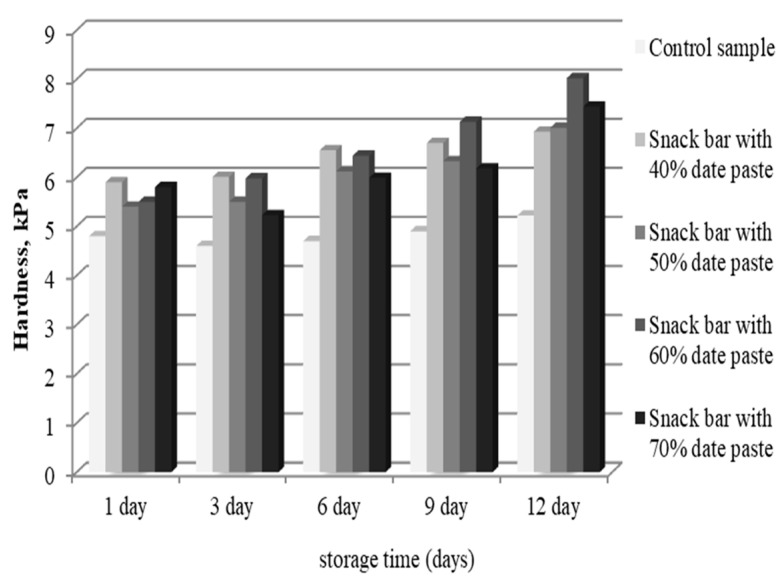
Changes in the hardness of five snack bar samples during 12-day storage.

**Table 1 foods-10-00918-t001:** Date bar formulations.

Ingredients	Control Sample	Date Bar with (%) Dates			
		40	50	60	70
Cashews, %	20.00	20.00	16.66	13.33	10.00
Oatmeal, %	20.00	20.00	16.66	13.33	10.00
Dried cherries %	20.00	20.00	16.66	13.33	10.00
Honey, %	40.00	-	-	-	-
Dates, %	-	40.00	50.00	60.00	70.00

**Table 2 foods-10-00918-t002:** Nutritional facts about the samples of snack bars with date fruit.

Nutritional Facts	Date Bar with (%) Dates				
	Control Sample	40	50	60	70
Energy, kcal	378.80	372.00	362.08	352.29	342.50
**Macronutrients**					
Protein, g	7.82	8.94	7.94	6.93	5.93
Fat, g	10.16	10.35	8.70	7.05	5.41
Carbohydrate, g	68.76	65.68	67.10	68.55	70.00
—Fibre, g	3.76	6.88	7.07	7.25	7.44
**Micronutrients**					
**Vitamins**					
Thiamin, mg	0.24	0.26	0.23	0.19	0.16
Riboflavin, mg	0.05	0.06	0.06	0.06	0.07
Niacin, mg	0.46	0.92	0.98	1.04	1.10
Pyridoxine, mg	0.11	0.17	0.17	0.17	0.17
Folate, μg	16.60	16.20	13.49	10.80	8.10
Pantothenic Acid, mg	0.45	0.43	0.36	0.29	0.22
**Minerals**					
Calcium, mg	29.70	102.10	116.24	130.40	144.55
Iron, mg	2.58	2.82	2.52	2.22	1.92
Magnesium, mg	94.60	153.80	153.14	152.52	151.90
Phosphorus, mg	224.8	234.00	199.43	164.96	130.50
Potassium, mg	238.60	408.32	419.58	430.94	442.31
Sodium, mg	4.40	5.80	6.08	6.37	6.65
Zinc, mg	2.04	2.07	1.77	1.48	1.18
Copper, mg	0.57	0.64	0.57	0.50	0.43
Manganese, mg	1.34	18.51	22.59	26.67	30.76
Selenium, μg	4.30	5.18	4.81	4.45	4.09

**Table 3 foods-10-00918-t003:** Sensory characteristics of snack bar samples.

Sensory Characteristics ^1^	Date Bar with (%) Dates				
	Control Sample	40	50	60	70
Overall acceptability	6.68 ^a^ ± 0.98	5.08 ^b^ ± 1.95	7.16 ^c^ ± 1.21	4.44 ^d^ ± 0.65	2.52 ^e^ ± 1.00
Appearance	5.92 ^a^ ± 0.95	5.88 ^a^ ± 1.05	7.28 ^b^ ± 0.89	6.96 ^c^ ± 0.93	6.36 ^d^ ± 1.07
Flavor	7.20 ^a^ ± 1.04	6.6 ^a^ ± 1.29	8.24 ^b^ ± 0.78	5.68 ^c^ ± 0.63	5.80 ^d^ ± 0.81
Sweetness	6.92 ^a^ ± 1.18	6.84 ^a^ ± 0.74	8.4 ^b^ ± 0.64	6.72 ^c^ ± 0.61	6.12 ^d^ ± 0.78
Texture	6.32 ^a^ ± 0.91	6.24 ^a^ ± 1.09	7.12 ^b^ ± 0.92	6.44 ^c^ ± 0.96	6.64 ^d^ ± 0.81

**^1^** Data are presented as means ± SD (standard deviation). ^a–e^ Means in a row not sharing the same capital superscript letter are significantly different at (*p* < 0.05, Tukey’s HSD test).

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
