# Peer review of "Application of Date (Phoenix dactylifera L.) Fruit in the Composition of a Novel Snack Bar"

_foods, 2021, doi:10.3390/foods10050918_

Round 1

Reviewer 1 Report

Dear Authors,

In my opinion, the paper is deeply revised, however in this step, you should pay particular attention to extend the discussion point in order to make it more detailed. The most crucial point is to compare the achieved results with the results available in the literature.

Author Response

Dear Reviewer,

Thank you for your letter and the opportunity to revise our paper. The suggestions offered by the reviewers have been immensely helpful, and we also appreciate your insightful comments on revising the abstract and other aspects of the paper.

We have included the reviewer comments immediately after this letter and responded to them individually, indicating exactly how we addressed each concern or problem and describing the changes we have made. The revisions have been approved by all authors. The changes are highlighted, using the "Track Changes" function as you requested, and the revised manuscript is attached to this email message.

We hope the revised manuscript will better suit the Foods journal and we thank you for your continued interest in our research.

Sincerely,

Hafize Fidan

Reviewer Comments, Author Responses and Manuscript Changes

Reviewer 1

Dear Authors, In my opinion, the paper is deeply revised, however in this step, you should pay particular attention to extend the discussion point in order to make it more detailed. The most crucial point is to compare the achieved results with the results available in the literature.

Response: Thank you for your valuable suggestions. We made the necessary revision and added more discussion.

We appreciate the opportunity to revise our work for consideration for publication in Foods. We hope our revision meets your approval. 

Reviewer 2 Report

  • I still don't understand how carbohydrates composition is higher than 100g / 100 g of product, specifically 102,83 and 139,98 g/100g in 60% and 70 % date bars.
  • as stated in the first revision, the results should follow the same order than the M&M section for the shake of consistency: aw, texture, nutritional information, etc. 

Author Response

Dear Reviewer,

Thank you for your letter and the opportunity to revise our paper. The suggestions offered by the reviewers have been immensely helpful, and we also appreciate your insightful comments on revising the abstract and other aspects of the paper.

We have included the reviewer comments immediately after this letter and responded to them individually, indicating exactly how we addressed each concern or problem and describing the changes we have made. The revisions have been approved by all authors. The changes are highlighted, using the "Track Changes" function as you requested, and the revised manuscript is attached to this email message.

We hope the revised manuscript will better suit the Foods journal and we thank you for your continued interest in our research.

Reviewer Comments, Author Responses and Manuscript Changes

Reviewer 2

  • I still don't understand how carbohydrates composition is higher than 100g / 100 g of product, specifically 102,83 and 139,98 g/100g in 60% and 70 % date bars.

Response: Thank you so much for catching that confusing error, which we have now corrected. The mistake was due to the technical error due to the differences in the date bars’ formulations.

  • as stated in the first revision, the results should follow the same order than the M&M section for the shake of consistency: aw, texture, nutritional information, etc. 
  • Response: Thank you! We have revised the order

We appreciate the opportunity to revise our work for consideration for publication in Foods. We hope our revision meets your approval.

Reviewer 3 Report

Need details of the raw material and where they were purchased. You just cannot write Dates were provided by Dr. Suleiman O. Aljaloud Where were these bars prepared?

How these bars were prepared/ just by mixing?

How sheeting was done?

Title say Novel Protein Snack Bar

What was the source of protein, only dates and cashew: Then we need specification for these ingredients?

In flow diagram they are talking about grinding but no details in preparation about grinding They also mentioned cooling. How and what was the time for cooling? Why cooling? Did they cook the mixture?

Author Response

Dear Reviewer,

Thank you for your letter and the opportunity to revise our paper. The suggestions offered by the reviewers have been immensely helpful, and we also appreciate your insightful comments on revising the abstract and other aspects of the paper.

We have included the reviewer comments immediately after this letter and responded to them individually, indicating exactly how we addressed each concern or problem and describing the changes we have made. The revisions have been approved by all authors. The changes are highlighted, using the "Track Changes" function as you requested, and the revised manuscript is attached to this email message.

We hope the revised manuscript will better suit the Foods journal and we thank you for your continued interest in our research.

Sincerely,

Hafize Fidan

Reviewer Comments, Author Responses and Manuscript Changes

Reviewer 3

Need details of the raw material and where they were purchased. You just cannot write Dates were provided by Dr. Suleiman O. Aljaloud Where were these bars prepared?

Response: Thank you for your suggestion. The text was revised to “Date (Phoenix dactylifera L.,) samples were provided by a National Organic Date company (Medjool brand) in Riyadh, the Kingdom of Saudi Arabia ), and standard raw materials such as honey (Region Strandzha, Bulgaria) cashews, oatmeal, and dried cherries (Vesta Trading LTD, Bulgaria) used in the current study were purchased from a local store in Plovdiv, Bulgaria and authorized by the Ministry of Health in Bulgaria”.

How these bars were prepared/ just by mixing?

Response: Thank you for your valuable comment. The text was revised to: “The samples were obtained in laboratory conditions in the technological laboratory of the University of Food Technology - Plovdiv, Bulgaria. The pulp of dates and dried cherries were separated from the seeds. The dates were ground into paste using a laboratory mill (Clatronic KSW 3307 Grinder) for 2-3 min. The other components - cashews and dried cherries are subjected to separate grinding with the mill. The oatmeal was used whole. The obtained date paste was then thoroughly mixed with other ingredients (oatmeals, dried cherries and cashews) in order to obtain uniform distribution. After mixing, sheeting was performed in order to create bars of 2.5 cm width, 1 cm height, and 7 cm in length. The date bars were obtained using a metal roller and polyethylene foil. The control sample was obtained with polyfloral honey; as in the other samples, it was replaced with date paste. The date bars are cooled (4 ± 6 ° C, for 30 minutes) in order to increase their hardness after being reduced due to the increased temperature from the mechanical impact of mixing. After being refrigerated, each date bar weighed approximately 25 g and were packed individually with polyethylene foil and plastic packaging. The samples thus prepared and coded were stored for 24 hours at room temperature (18 ± 2 ° C) before their physical and sensory testing. The formulations of four date bars (Figure 2) containing date paste, cashews, dried cherries and oatmeal are given in Table 1, with ingredient amounts being shown as percentages. The control sample formulation was prepared according to traditional techniques as the ingredients (date paste or honey, oatmeal, dried cherries and cashews) were mixed”.

How sheeting was done?

Response: After mixing, sheeting was performed in order to create bars of 2.5 cm width, 1 cm height, and 7 cm in length. The sheeting was made using a metal roller and polyethylene foil.

Title say Novel Protein Snack Bar

Response: Thank you! The title was revised to “Novel Snack Bar”

What was the source of protein, only dates and cashew: Then we need specification for these ingredients?

Response: Thank you so much for your suggestion. We have gone through the entire manuscript carefully and made the necessary corrections in order to make the specification more precise, as we changed the term “protein bar” to “date bar”.

In flow diagram they are talking about grinding but no details in preparation about grinding They also mentioned cooling. How and what was the time for cooling? Why cooling? Did they cook the mixture?

Response: Thank you for your valuable comment. The flow diagram was removed because the procedure is described more precisely. The date bars are cooled (4 ± 6 ° C, for 30 minutes) in order to increase their hardness after being reduced due to the increased temperature from the mechanical impact of mixing. After being refrigerated, each date bar weighed approximately 25 g and were packed individually with polyethylene foil and plastic packaging. The samples thus prepared and coded were stored for 24 hours at room temperature (18 ± 2 ° C) before their physical and sensory testing. The mixture was not cooked.

Round 2

Reviewer 3 Report

They made most of the correction 

This manuscript is a resubmission of an earlier submission. The following is a list of the peer review reports and author responses from that submission.

Round 1

Reviewer 1 Report

Although applying dates as a substitute for honey in snack bar is interesting, the way the study is carried out, unfortunately, does not fit the standard for foods.

The sensory analysis portion of the method is critically ill-conducted. Especially the selection of the panel was not representative of the general snack bar users. The results and interpretation of the results can be very misleading. 

Here are my comments.

1. The abstract can be more concise.

2. In the Introduction "A number of misconceptions regarding food and nutrition have developed over the last century leading to the emergence of products that are attractive in appearance but low in nutritional value."

What kind of misconception??? the description is too general and the meaning does not deliver precisely. So much things happened and I am not sure your statement supports the current obesity and health problems.

3. I think the description of date’s nutrient content can be shorten since pulp is the relevant part in the present study. Might as well talk more about processed food using dates and their market size and potential growth. Just substituting dates with honey due to lack of allergens is not that convincing. Please add more justification of why date has to be applied to protein bars.

4. In the Materials and Methods, The authors have to write out not only the brand of the material and producer and country.

5. In the sample information, the ingredients do not add upto a hundred. For example, Control, it is 110%, other samples they are 84% to 102%. How should the readers interpret these? The overall proportion add up to a hundred. How did the authors fill the rest of the missing mass for sample like 40% dates. There are 26% gap with control. How did you make up the rest of the portion?

6. You do not determine the acceptance level with trained panel. They are not qualified to evaluate the liking of snack bars. Naïve consumers or untrained consumers who enjoy consuming snack bar should be recruited. Usually the number of consumers necessary is 60+

Reviewer 2 Report

The topic of the study is important and interesting. In general, the procedure, the study design and statistical analyses are well organized.

The paper is easy to read for the final reader. However, I would like to indicate some major suggestions to authors.

Introduction

In my opinion the Authors should indicate more references in this part in order to present the current situation referring to this topic.

Material and Methods (sensorial evaluation)

Could you please explain some points:

  • The age of the panelists (the range, the mean).
  • How do you reduce the possible order effect?
  • How many sessions were used; how many replications?
  • What kind of methods you used in order to minimize the error and masking of sensory attributes.
  • You used the trained panelists in your study in order to evaluate the samples. Did you also use the consumer assessment in this study (I think about consumers who are buyers of this kind of the snack bars/bars with nutritional value)?

Conclusions

Are there any additional practical implications of your findings?

Reviewer 3 Report

  • One of the main concerns is about the word “protein”. The tittle, keywords, and introduction include the word and the concept “protein”. Is the date bar a protein bar? It includes about 10% protein and a lot of carbohydrate….I am not sure it is a protein bar as those in the market elaborated with soy protein; it seems more likely a carbohydrate-based bar.
  • Other issue is related with the formulation. In figure 1 the authors are stating this is a corn syrup substitution. However, the authors are not substituting corn syrup but honey. This is difficult to understand. In fact, they are elaborating a new and different bar; the formulation in this work has little to do with the commercial example given in Figure 1. The aim of the manuscript isn’t clear (lines 71-73), nothing about substitution is said here. Are the authors using date palm for substitution of syrup, honey, no substitution is made???
  • The tables are difficult to understand. What is “date weight %”? (table 1) If the authors are giving the rest of ingredients in relation to date palm weight, the value for date should be 100. In table 2, why comparing (statistics) within the same column? Is overall acceptability, being compared to appearance and with texture and so on?. They cannot be compared. In the case of table 3, why control is not included? How can the bars have 139,98 g carbohydrate/ 100 g bar????? Moreover, mcg is not a correct unit. The unit symbol is μg according to the International System of Units. Table 4 should be shortened indicating only the detected microorganisms.
  • The quality of the figures should be improved. Moreover, figure 2 is not needed. It can be explained in the text. Figure 1 could be deleted as well as the “substitution” is not clear and the commercial example given has little to do with the control elaborated in the work.
  • The sensory approach is not appropriate. For hedonic tests, consumers ( at least 80-100 consumers) should be used instead of trained panelists. See ISO 11136-2014.
  • The choice of double compression for texture analysis is not appropriate. A penetration test or a cutting test should be selected for this type of products
  • The results should follow the same order than the M&M section for the shake of consistency.
  • Statistics should be used more precisely. E.g. line 165. If no statistical differences are detected, you cannot state " approximately" you should state that " no significant differences were observed".